# Common and rare genetic variants predisposing females to unexplained recurrent pregnancy loss

Kyuto Sonehara [1,2,3,33], Yoshitaka Yano[4,33], Tatsuhiko Naito [2,3], Shinobu Goto[4], Hiroyuki Yoshihara [4], Takahiro Otani[5], Fumiko Ozawa[4], Tamao Kitaori[4], the Biobank Japan Project*, Koichi Matsuda [6,7], Takashi Nishiyama[5], Yukinori Okada [1,2,3,8,9,34] ✉ & Mayumi Sugiura-Ogasawara [4,34] ✉

Recurrent pregnancy loss (RPL) is a major reproductive health issue with multifactorial causes, affecting 2.6% of all pregnancies worldwide. Nearly half of the RPL cases lack clinically identifiable causes (e.g., antiphospholipid syndrome, uterine anomalies, and parental chromosomal abnormalities), referred to as unexplained RPL (uRPL). Here, we perform a genome-wide association study focusing on uRPL in 1,728 cases and 24,315 female controls of Japanese ancestry. We detect significant associations in the major histocompatibility complex (MHC) region at 6p21 (lead variant=rs9263738; $P = 1.4 \times 10^{-10}$; odds ratio [OR] = 1.51 [95% CI: 1.33−1.72]; risk allele frequency = 0.871). The MHC associations are fine-mapped to the classical HLA alleles, HLA-C*12:02, HLA-B*52:01, and HLA-DRB1*15:02 ($P = 1.1 \times 10^{-10}$, $1.5 \times 10^{-10}$, and $1.2 \times 10^{-9}$, respectively), which constitute a population-specific common long-range haplotype with a protective effect ($P = 2.8 \times 10^{-10}$; OR = 0.65 [95% CI: 0.57−0.75]; haplotype frequency=0.108). Genome-wide copy-number variation (CNV) calling demonstrates rare predicted loss-of-function (pLoF) variants of the cadherin-11 gene (CDH11) conferring the risk of uRPL ($P = 1.3 \times 10^{-4}$; OR = 3.29 [95% CI: 1.78−5.76]). Our study highlights the importance of reproductive immunology and rare variants in the uRPL etiology.

Recurrent pregnancy loss (RPL), also referred to as recurrent miscarriage, is a major issue in reproductive medicine, defined as the occurrence of two or more spontaneous losses of pregnancy[1–3]. The prevalence of RPL was reported to be approximately 2.6% of all pregnancies worldwide, and most of them occur in the first trimester[1]. Maternal age at conception is a strong risk factor for miscarriage. In developed countries, where the female age at pregnancy has elevated year by year recently, the prevalence of RPL is increasing (e.g., 5.0% in Japan), becoming a topic of growing importance in reproductive health.

The underlying etiology of RPL is considered highly heterogeneous, involving immunologic, anatomic, cytogenetic, endocrinological, and infectious factors[4]. Clinically identifiable causes of RPL include antiphospholipid syndrome (APS), uterine anatomic anomalies, and parental chromosomal abnormalities[1–3,5]. While these established causes account for nearly half of RPL cases, the remaining half show no apparent clinical causes, referred to as unexplained RPL (uRPL). The lack of etiological explanations poses a tough challenge for clinical management and exacerbates the patients' psychological distress. uRPL can be classified into aneuploid and euploid miscarriage according to the embryonic karyotype by examining their aborted products[6]. Aneuploid and euploid miscarriage are different in

---

A full list of affiliations appears at the end of the paper. *A list of authors and their affiliations appears at the end of the paper.
✉e-mail: yuki-okada@m.u-tokyo.ac.jp; og.mym@med.nagoya-cu.ac.jp

their epidemiological characteristics and the cumulative live birth rate[7], indicating distinct underlying biology.

Genetic determinants of uRPL are a major subject of interest for elucidating the underlying etiology. A number of articles concerning uRPL genetics have been published after an association of the C677T variant in the methylenetetrahydrofolate reductase (*MTHFR*) gene was reported[8]. Susceptibility genes implicated to date include those involved in immune response, coagulation, metabolism, and angiogenesis. However, most of the studies employed a candidate gene approach with a limited sample size ($n < 200$) and varied case definitions. A recent systematic review carried out a meta-analysis of these studies and demonstrated predominantly inconsistent results, warranting hypothesis-free study design with a large sample size[9].

Recently, Laisk et al. carried out a European ancestry genome-wide association meta-analysis of RPL with the case defined as having a history of three or more consecutive miscarriages ($n_{case} = 750$), reporting three genome-wide significant loci[10]. No association was ascertained between RPL and variants which were reported significant in the aforementioned systematic review[9]. This report was based on the existing biobank data, and most of the studied cases were not examined for the clinically identifiable causes of RPL and were not characterized for relevant clinical features, such as embryonic karyotypes. Given that the clinical heterogeneity in the studied cases potentially masked susceptibility loci responsible for uRPL, genome-wide association studies (GWAS) with detailed phenotyping of the cases are required for uncovering the underlying etiological factors of uRPL.

Here, by focusing on uRPL cases clinically ascertained not to have conventional causes, we perform the largest GWAS of uRPL to date, accompanied by stratified analyses according to clinical features, to provide insights into the genetic basis of uRPL. Our data reveal an association between uRPL and the major histocompatibility complex region (MHC), which is fine-mapped to specific HLA alleles by an HLA imputation analysis. Finally, given the strong purifying selection pressure imposed on the uRPL risk alleles, we conduct genome-wide rare copy-number variation (CNV) calling and explore its contribution to the uRPL predisposition.

## Results
### Study participants
In this study, a total of 1800 patients with uRPL and 25,999 female controls were enrolled and genotyped with the use of Illumina Infinium Asian Screening Array. During patient enrollment, those with known causes of RPL (i.e., APS, uterine anomalies, and parental chromosomal abnormalities) were exhaustively excluded from the analysis through systematic clinical, cytogenetic, and serological examinations (Methods). After stringent quality control (QC) filters were applied to the genotyped data, 1728 uRPL cases and 24,315 controls, both of Japanese ancestry, were retained for the genetic association analysis. The demographic characteristics are summarized in Table 1. The age at the first visit and number of pregnancy losses of the cases were $34.1 \pm 4.7$ years (range, 19–48 years) and $2.7 \pm 0.95$ times (range, 2–12 times), respectively. The 1728 uRPL cases contained 843 patients with a history of three or more pregnancy losses (48.8%). The embryonic karyotypes of abortus were examined and classified into the following six categories (Methods): euploid miscarriage ($n = 204$, 11.8%); aneuploid ($n = 125$, 7.2%); 45,X ($n = 46$, 2.7%); triploid ($n = 31$, 1.8%); other abnormalities ($n = 30$, 1.7%); and unknown miscarriage ($n = 1292$, 74.8%). Antinuclear antibodies (ANA) titer in the serum was measured for 1635 uRPL cases, 543 of which were positive (≥1:40). Free T4 level was measured for 1621 uRPL cases, 285 of which showed a low value (≤0.9). Hundred and forty-four patients with uRPL had a past medical history of autoimmune diseases, including systemic lupus erythematosus ($n = 1$), rheumatoid arthritis ($n = 5$), chronic thyroiditis ($n = 98$), and other diseases ($n = 40$).

### Genome-wide association study
After whole-genome genotype imputation using a population-specific reference panel ($n = 4561$), we obtained 8,717,431 autosomal and X-chromosome variants fulfilling the post-imputation QC criteria (i.e., minor allele frequency [MAF] > 0.5% and $Rsq > 0.7$). Based on the imputed genotype dosages, we performed a single-variant GWAS of uRPL. To robustly control for population stratification and sample relatedness, we employed a generalized linear mixed model (GLMM) implemented in the SAIGE[11] software for the association test. The overall distribution of the association statistics did not show systematic inflation due to potential confounding biases such as residual population stratification (genomic inflation factor [$\lambda_{GC}$] = 1.026; linkage disequilibrium [LD] score regression intercept = $0.992 \pm 0.010$; the quantile–quantile plot is shown in Supplementary Fig. 1). The liability scale heritability of uRPL estimated by LD score regression was $0.307 \pm 0.143$. We observed the MHC region at 6p21 surpassing the genome-wide significance threshold (the smallest $P = 1.4 \times 10^{-10}$; Fig. 1; Table 2). The lead variant rs9263738 has two alleles of T/C, and the T allele showed a susceptible effect on uRPL (odds ratio [OR] = 1.51; 95% confidence interval [CI] = 1.33–1.72). We note that the C677T variant in the *MTHFR* gene previously implicated with uRPL (rs1801133)[8] did not reach nominal significance in our GWAS ($P = 0.12$; OR = 1.06; 95% CI = 0.98–1.14).

In our primary GWAS above, the control group included all the female participants available in the Biobank Japan project (BBJ) to maximize statistical power for detecting susceptibility loci. However, given the hospital-based cohort design of BBJ, our primary GWAS could have the following potential limitations which may introduce false positive associations: i) high prevalence for other MHC-associated diseases in the controls and ii) difference in the body mass index (BMI), a known risk factor for RPL[12], between cases and controls. To address these potential limitations, we additionally performed a sensitivity analysis. We confined the controls to 9,955 individuals who had a definitive record of the ICD10 code and were explicitly free of Crohn's disease (ICD10 K50), ulcerative colitis (K51), and antiphospholipid syndrome (D686). To account for the BMI difference, we estimated age-matched BMI based on the relationship between age and BMI (Methods). When we repeated the association test using the strictly defined control set with adjustment for the age-matched BMI, the lead variant rs9263738 remained genome-wide significant with comparable effect size ($P = 7.5 \times 10^{-10}$; OR = 1.52 [95% CI = 1.33–1.74]), indicating the robustness of the association between uRPL and MHC.

### MHC fine-mapping analysis
To fine-map the significant association within the MHC region, we performed an HLA imputation analysis using a high-resolution reference panel of 1118 Japanese individuals[13] (Methods). Applying the post-imputation quality control filter (MAF > 0.5% and $r^2$ by DEEP*HLA > 0.7), we obtained genotype dosages of 108 two-digit, 164 four-digit, and 173 six-digit HLA alleles, as well as 1666 amino acid polymorphisms of classical and nonclassical HLA genes in the entire MHC region. When evaluating the association of the imputed HLA variants with the risk of uRPL based on the strictly defined control set, we observed the most significant signal at *HLA-C* (Fig. 2a; Supplementary Data 1). The lead *HLA-C* alleles were HLA-C\*12:02:02 ($P = 4.8 \times 10^{-11}$; OR = 0.66; 95% CI = 0.58–0.75) and its four-digit allele, HLA-C\*12:02 ($P = 1.1 \times 10^{-10}$; Table 3). Notably, HLA-C\*12:02 constitutes a Japanese-specific common long-range haplotype spanning the entire HLA class I and class II regions (HLA-C\*12:02–HLA-B\*52:01–HLA-DRB1\*15:02)[14,15], which has a susceptible effect on ulcerative colitis but a protective effect on Crohn's disease[14]. We note that the association at *HLA-C* showed the most significant signal when we performed the analysis with all the available control individuals (i.e., the controls with a known history of ulcerative colitis or Crohn's disease were not explicitly excluded). In

**Table 1 | Characteristics of study participants ($n$ = 1728 cases and 24,315 controls)[a]**

| Characteristics | Case n | (%) | Early miscarriage | Intrauterine fetal death | Chrioamnionitis | Control n | (%) |
|---|---|---|---|---|---|---|---|
| *Age at sampling* | | | | | | | |
| <20 | 1 | 0.0 | | | | 115 | 0.5 |
| 20–29 | 329 | 19.0 | | | | 335 | 1.4 |
| 30–39 | 1158 | 67.0 | | | | 1488 | 6.1 |
| ≥40 | 240 | 13.9 | | | | 22,377 | 92.0 |
| *Body Mass Index* | | | | | | | |
| <18.5 | 272 | 15.7 | | | | 3405 | 14.0 |
| 18.5-24.9 | 1275 | 73.8 | | | | 15,882 | 65.3 |
| ≥25 | 163 | 9.4 | | | | 4471 | 18.4 |
| Missing | 18 | 1.0 | | | | 557 | 2.3 |
| *Number of previous pregnancy loss* | | | | | | | |
| 0 | 0 | 0.0 | 0 | 0 | 0 | | |
| 2 | 884[b] | 51.1 | 864 | 71 | 25 | | |
| 3 | 586 | 33.9 | 580 | 62 | 22 | | |
| ≥4 | 257 | 14.9 | 257 | 38 | 17 | | |
| *Number of previous live births* | | | | | | | |
| 0 | 1330 | 77.0 | | | | | |
| ≥1 | 398 | 23.0 | | | | | |
| *Embryonic (fetal) karyotype* | | | | | | | |
| Euploid | 204 | 11.8 | | | | | |
| Aneuploid | 125 | 7.2 | | | | | |
| 45,X | 46 | 2.7 | | | | | |
| Triploid | 31 | 2.4 | | | | | |
| Others | 30 | 1.7 | | | | | |
| Unknown | 1292 | 74.8 | | | | | |
| *Antinuclear antibody* | | | | | | | |
| Negative | 1092 | 63.2 | | | | | |
| Positive (≥1:40) | 543 | 31.4 | | | | | |
| Missing | 93 | 5.4 | | | | | |
| *Hypothyroidism* | | | | | | | |
| Presence free T4 ≤ 0.9 | 285 | 16.5 | | | | | |
| Absence | 1336 | 77.3 | | | | | |
| Missing | 107 | 6.2 | | | | | |
| *Autoimmune disease* | | | | | | | |
| Systemic lupus erythematosus | 1 | 0.1 | | | | | |
| Rheumatoid arthritis | 5 | 0.3 | | | | | |
| Chronic thyroiditis | 98 | 5.7 | | | | | |
| Absence | 1584 | 91.7 | | | | | |
| Others | 40 | 2.3 | | | | | |

[a]Including 65 cases diagnosed as weakly positive for antiphospholipid antibody.
[b]Including 2 cases in which the patient was diagnosed with a history of two miscarriages, but one of the two miscarriages was later diagnosed as a chemical pregnancy.

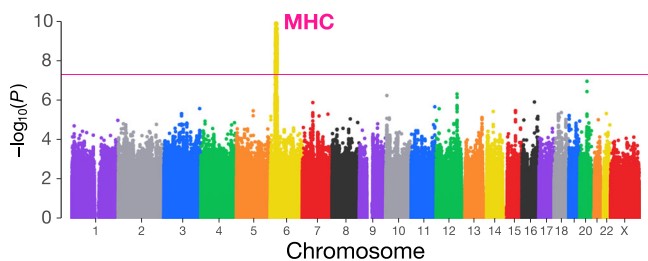

**Fig. 1 | Genome-wide association study of unexplained recurrent pregnancy loss.** Genome-wide associations of imputed genetic variants are shown. The pink horizontal line indicates the genome-wide significance threshold of $P = 5.0 \times 10^{-8}$. $P$-values were computed using SAIGE. All statistical tests are two-sided and unadjusted for multiple comparisons. MHC, major histocompatibility complex.

line with the earlier studies, the other HLA alleles constituting the long-range haplotype, HLA-B*52:01 and HLA-DRB1*15:02, showed comparable associations ($P = 1.5 \times 10^{-10}$ and $P = 1.2 \times 10^{-9}$, respectively; Fig. 2a; Table 3). When conditioned on any single HLA allele of HLA-C*12:02, HLA-B*52:01, and HLA-DRB1*15:02, no variants in the MHC region reached the significance threshold (Fig. 2b,c,d). Motivated by this observation, we further performed genotype imputation of the long-range haplotype of HLA-C*12:02–HLA-B*52:01–HLA-DRB1*15:02 (Methods) and evaluated its association with uRPL. As expected, HLA-C*12:02–HLA-B*52:01–HLA-DRB1*15:02 haplotype showed a significant protective effect ($P = 2.8 \times 10^{-10}$; OR = 0.65; 95% CI = 0.57–0.75; Table 3). When conditioned on HLA-C*12:02–HLA-B*52:01–HLA-DRB1*15:02, no significant association was observed in the MHC region (Fig. 2e).

**Table 2 | Association summary of the lead genetic variant**

| rsID | Chr | Position (hg19) | Alleles | Risk Allele | Risk allele frequency | | OR (95% CI) | *P*-value |
|---|---|---|---|---|---|---|---|---|
| | | | | | Case | Control | | |
| rs9263738 | 6 | 31,109,767 | T/C | T | 0.893 | 0.871 | 1.51 (1.33–1.72) | $1.4 \times 10^{-10}$ |

Two-sided association testing was performed using SAIGE. The *P*-value is unadjusted for multiple comparisons.

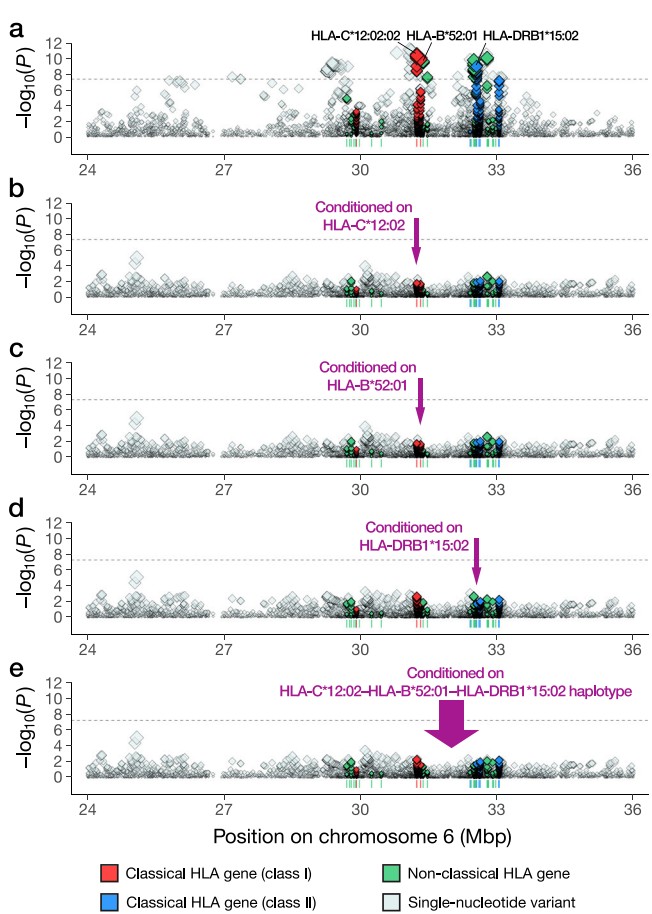

**Fig. 2 | MHC fine-mapping analysis on the association with uRPL.** Regional associations of the imputed HLA variants in the MHC region with uRPL are shown. **a** Nominal regional associations. **b** Regional associations conditioned on HLA-C*12:02. **c** Regional associations conditioned on HLA-B*52:01. **d** Regional associations conditioned on HLA-DRB1*15:02. **e** Regional associations conditioned on HLA-C*12:02–HLA-B*52:01–HLA-DRB1*15:02 haplotype. Each diamond represents the −log₁₀(*P*) of the variants, including the single-nucleotide variants, two-, four-, and six-digit HLA alleles, and amino acid variants of HLA genes. The horizontal dashed line represents the genome-wide significance threshold of $P = 5.0 \times 10^{-8}$. Detailed association results are available in Supplementary Data 1. *P*-values were computed by logistic regression. All statistical tests are two-sided and unadjusted for multiple comparisons.

## Stratified analysis according to clinical features

Given the heterogeneity in the underlying biology of uRPL, we stratified the GWAS participants according to clinical features, such as the embryonic karyotype and immunological characteristics. We assumed that uRPL with different embryonic karyotypes involves different etiologies; in particular, abnormal embryonic karyotypes would in itself serve as a predominant cause of miscarriage[7]. To reduce heterogeneity in the etiological background of the uRPL cases, we excluded the cases showing embryonic karyotype abnormalities, defining the remaining 1480

cases as case group (A) (Supplementary Fig. 2a). Motivated by the significant association at MHC, the genetic locus well known to be involved in immune function, we further stratified case group (A) based on the immunological features. Specifically, we defined 459 cases with positive ANA as case group (B) (Supplementary Fig. 2b). High ANA titer is commonly detected in patients with RPL, debated to be involved in the unexplained part of the disease etiology. Given that the MHC region is involved in a wide range of autoimmune diseases susceptibility, we hypothesized that the association between MHC and uRPL may be mediated by ANA production. If so, restricting the GWAS cases to those with positive ANA would result in a substantially larger effect size of the lead HLA haplotype HLA-C*12:02–HLA-B*52:01–HLA-DRB1*15:02. Despite this expectation, the odds ratio was nearly equivalent whether limiting to positive ANA or not (Fig. 3). We note that this observation does not refute ANA involvement in the etiology of uRPL, since ANA may mediate uRPL pathogenesis independent of the MHC locus. Next, by excluding the cases with positive ANA, past medical history of autoimmune diseases, and hypothyroidism, we defined the remaining 694 cases as case group (C). When comparing the case group (C) with the controls, the HLA association remained significant, indicating that the HLA association did not originate from the risk alleles of the known autoimmunity (Fig. 3; Supplementary Fig. 2c). These results collectively suggest that the MHC association is independent of autoantibody presence. No genome-wide significant association was observed outside the MHC region in the GWAS of the case group (A), (B), or (C) (Supplementary Fig. 2a,b,c). We note that case groups (A), (B), and (C) contained uRPL cases with unknown karyotypes to retain sample size for the stratified analysis. To focus exclusively on euploid and aneuploid miscarriage, we defined the 203 cases with confirmed embryonic euploidy as case group (D) and the 124 cases with confirmed embryonic aneuploidy as case group (E). We performed a GWAS of case group (D) and (E), finding no genome-wide significant association (Supplementary Fig. 2d,e). In these settings, HLA-C*12:02–HLA-B*52:01–HLA-DRB1*15:02 haplotype did not reach statistical significance due to the reduced sample size (*P* = 0.16 for (D) and *P* = 0.30 for (E); Fig. 3).

## Rare copy-number variation associated with uRPL

From an evolutionary perspective, given that uRPL is directly related to reproductive fitness, risk alleles of uRPL should be exposed to particularly strong purifying selection pressure. Rare functional genetic variants are expected to be a key heritable component of uRPL susceptibility. Among rare genetic variants, a rare CNV affects a larger fraction of the genome sequence (i.e., >50 bp) than a single-nucleotide variant and short insertion/deletion, thus presumed to have a strong functional effect. To investigate the contribution of rare functional CNVs to uRPL susceptibility, we performed genome-wide rare CNV calling using the HI-CNV software, which leverages haplotype-sharing information in the biobank-scale data to increase the sensitivity for CNV detection. We detected a median of 12 CNV calls per individual for both the case and control group (Supplementary Fig. 3). To identify the genes of which deleterious CNV burden contributes to uRPL, we compared the number of predicted loss-of-function (pLoF) CNV

**Table 3 | Association summary of the lead HLA alleles and haplotype**

| Allele/Haplotype | Frequency | | OR (95% CI) | P-value |
|---|---|---|---|---|
| | Case | Control | | |
| HLA-C*12:02 | 0.105 | 0.128 | 0.67 (0.59–0.76) | $1.1 \times 10^{-10}$ |
| HLA-B*52:01 | 0.105 | 0.127 | 0.67 (0.59–0.76) | $1.5 \times 10^{-10}$ |
| HLA-DRB1*15:02 | 0.0968 | 0.118 | 0.67 (0.59–0.77) | $1.2 \times 10^{-9}$ |
| HLA-C*12:02–HLA-B*52:01–HLA-DRB1*15:02 | 0.0868 | 0.108 | 0.65 (0.57–0.75) | $2.8 \times 10^{-10}$ |

Two-sided association testing was performed using logistic regression. *P*-values are unadjusted for multiple comparisons.

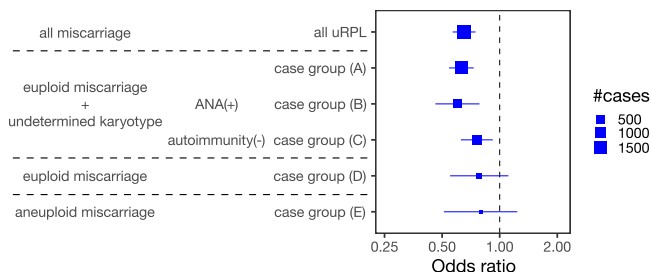

**Fig. 3 | Associations of HLA-C*12:02–HLA-B*52:01–HLA-DRB1*15:02 haplotype in the stratified analysis.** Odds ratios of HLA-C*12:02–HLA-B*52:01–HLA-DRB1*15:02 haplotype in the stratified analysis are shown. The marker size is proportional to the case sample size. Error bars indicate 95% CI, and the measure of center is the maximum likelihood estimate by logistic regression. case group (A), uRPL cases with confirmed embryonic euploid or undetermined karyotype; case group (B), ANA(+) uRPL cases with confirmed embryonic euploid or undetermined karyotype; case group (C), autoantibody and hypothyroidism(-) uRPL cases with confirmed embryonic euploid or undetermined karyotype; case group (D), uRPL cases with confirmed embryonic euploid; case group (E), uRPL cases with confirmed embryonic aneuploid.

carriers between uRPL cases and controls (Methods). Rare pLoF CNVs of *CDH11* (carried by 0.93% of the cases and 0.28% of the controls) showed a significant association with the increased risk of uRPL after Bonferroni correction ($P = 1.3 \times 10^{-4}$; OR = 3.29; 95%CI = 1.78–5.76; Fig. 4a; Supplementary Data 2). All the *CDH11* pLoF CNVs detected were deletions (Fig. 4b), and the carriers in the cases and controls were all heterozygous. *CDH11* encodes cadherin-11, which is prominently expressed in the female reproductive system according to the Human Protein Atlas database[16] (Supplementary Fig. 4) and plays a vital role in the differentiation and fusion of trophoblastic cells in vitro[17].

## Discussion

In this work, we performed a GWAS of uRPL with the largest case sample size ever reported, identifying genome-wide significant associations in the MHC region. The association was fine-mapped to a population-specific HLA haplotype of HLA-C*12:02–HLA-B*52:01–HLA-DRB1*15:02, which is previously reported to increase the risk of ulcerative colitis and decrease the risk of Crohn's disease. Furthermore, we interrogated the contribution of rare CNVs to the uRPL risk, revealing a significantly high pLoF CNV burden of *CDH11* in patients with uRPL.

The HLA genes play a critical role in adaptive immune responses and maintenance of self-tolerance. Despite the fact that the fetus is a semi-allograft for the mother, it escapes immunological rejection. As pointed out by Medawar as 'immunological paradox of pregnancy' in 1953, pregnancy is an immunologically unique time when two genetically distinct individuals coexist[18]. Extravillous trophoblasts (EVT), the invasive form of differentiated trophoblast cells in direct contact with all maternal decidual cells, express a unique combination of MHC molecules, including HLA-C, E, and G antigens but not HLA-A, HLA-B,

or class II antigens[19]. Since HLA-E and HLA-G have limited genetic variation in human populations, the polymorphism in the HLA-C alleles has been regarded as a major genetic determinant of allorecognition in the mother. The HLA-C allotypes are recognized by uterine natural killer cell (uNK), a key immune cell type for pregnancy accounting for 70% of decidual leukocytes in the first trimester[20]. HLA-C variants have been implicated in pregnancy disorders, including pre-eclampsia and high birthweights, via the uNK allorecognition system[21,22]. Our data provide the first evidence of the association between HLA polymorphism and RPL with genome-wide significance, corroborating the importance of immunological tolerance in a successful pregnancy.

High ANA titer and hypothyroidism (generally accompanied by anti-thyroid peroxidase antibody) are commonly observed in patients with RPL; however, their presence does not serve as a predictor of subsequent miscarriages, and their role in RPL pathogenesis is controversial[5,23,24]. In the stratified analysis, we demonstrated that the MHC association was consistently observed independent of the autoantibody presence in the cases (Fig. 3). Our data suggests that the MHC association is mediated by cell-mediated immunity rather than antibody-mediated immunity, which is also in line with the putative involvement of the uNK allorecognition system discussed above.

The MHC association was unreported in the previous GWAS of RPL in the European population[10]. One probable explanation for the discrepancy is the difference in the case definition. The GWAS cases in the previous study include those with established causes, which may result in attenuation of the association signal for unexplained RPL. Another probable explanation is the difference in allele frequency between the populations. The associated HLA alleles are rare variants in the European population (e.g., allele frequency = 0.0038 for HLA-C*12:02, HLA-B*52:01, and HLA-DRB1*15:02 according to the 1000 Genomes project European population[25]). The low allele frequency in European populations may limit statistical power for detecting the association. We note that the population-specificity and long-range LD of the protective HLA haplotype suggests positive selection due to enhanced reproductive success. The details of the positive selection process and the effect of this HLA haplotype on other traits may be further investigated as potential future research.

The comparative analysis of gene-damaging CNV burden between uRPL cases and controls nominated *CDH11*, one of the type 2 classical cadherins from the cadherin superfamily that mediates calcium-dependent cell-cell adhesion. *CDH11* is expressed in the epithelium of the placenta as well as endometrial stroma, supposed to play a role in anchoring trophoblasts to the decidua[26]. The expression of *CDH11* promotes differentiation and fusion of cytotrophoblasts to form syncytiotrophoblasts in vitro[17]. The damaged *CDH11* coding sequence in the mother is inherited by the fetus with a 50% chance; thus, the pLoF CNVs potentially confer the disease risk by impairing the physiological function of the fetal tissues, including trophoblasts, although the direct target of our analysis is the maternal genomes.

Our study has some potential limitations. First, the control participants for the association analyses were derived from the existing biobank, and not all of them were confirmed to be free of uRPL. However, the potential misclassification of the controls in our GWAS

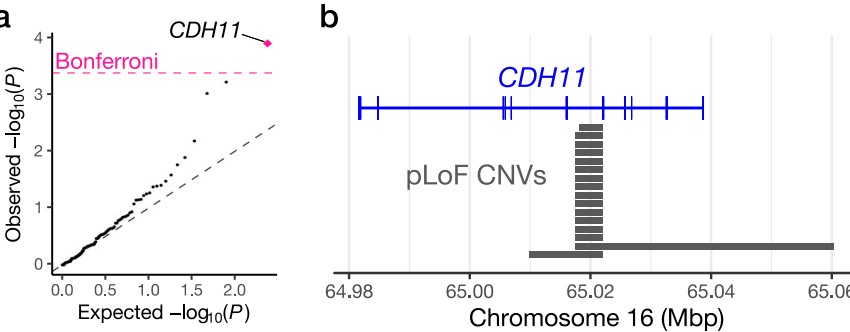

**Fig. 4 | Predicted loss-of-function copy-number variation associated with uRPL. a** A quantile–quantile plot of the *P*-values of tested genes from the pLoF CNV burden test. The gray diagonal dashed line is the identity line. The pink horizontal dashed line indicates the Bonferroni-corrected significance threshold of α = 0.05.

Detailed association results are available in Supplementary Data 2. All statistical tests are two-sided and unadjusted for multiple comparisons. **b** Locations of the *CDH11* pLoF CNVs in the uRPL cases are shown.

does not undermine the robustness of our findings. Even if some controls had a history of pregnancy loss, it would not lead to false positive associations but rather conservative results with underestimated heritability and odds ratio. Second, although we performed the sensitivity analysis to account for the patients of MHC-associated diseases in the controls, the highly pleiotropic nature of the MHC locus could affect the association signal by other MHC-associated diseases potentially enriched in the controls. Third, in the sensitivity analysis, we adjusted for BMI by modeling the BMI trajectory as a function of age. Our model was relatively simple and may not fully capture the potential difference in the BMI difference at reproductive age between cases and controls. Last, the rare CNVs at *CDH11* were detected using an SNP array intensity-based method. Since the SNP array probes are designed to be distributed genome-wide at intervals of more than 5 kbp, locating the precise genomic coordinates at which the variants start or end involves technical challenges. We also note another possibility that the rare variant association potentially represents more complex structural variants rather than simple deletions.

Collectively, we conducted a large-scale GWAS of uRPL and revealed the significant contribution of the HLA polymorphism to the disease predisposition. Through a genome-wide rare CNV analysis, we also demonstrated that deleterious rare variants confer the risk of uRPL. Our findings should shed light on the key role of reproductive immunology and rare genetic variants in the currently unexplained etiology of uRPL.

## Methods
### Study population
We enrolled 1800 Japanese patients with a history of two or more unexplained pregnancy losses. All patients were recruited from Nagoya City University Hospital between May 2007 and July 2022. All medical information, including the history of pregnancy losses, was obtained through medical interviews by the obstetricians. Chemical pregnancies were not included in pregnancy losses. All patients underwent a systematic examination, including 3D-ultrasound sonography, chromosome analysis of both partners, determination of antiphospholipid antibody, including lupus anticoagulant, by diluted activated partial prothrombin time, diluted Russell viper venom time and β2 glycoprotein I-dependent anticardiolipin antibody, and blood tests for hypothyroidism and diabetes mellitus, before a subsequent pregnancy. Patients with APS, an abnormal chromosome in either partner, or uterine anomaly were excluded. When a missed miscarriage was diagnosed, a dilatation and curettage or manual vacuum aspiration was performed and cytogenetic analysis of products of conception was carried out. Subsequent pregnancy outcomes were followed up until May 2023 by a review of the medical records. Embryonic

karyotypes were primarily examined in the latest pregnancy loss for the patients. Some cases were examined multiple times, and if both euploid and aneuploid were observed in a case, they were classified into aneuploid miscarriage. As the control population, DNA samples of 26,037 females were obtained from the Biobank Japan Project (BBJ)[27,28]. This study was conducted with the approval of the Research Ethics Committee of Nagoya City University Graduate School of Medical Sciences, the University of Tokyo, and Osaka University. All participants provided written informed consent after being given a full explanation of the purpose of the study and the methods to be employed. This study complies with the Declaration of Helsinki.

### Genotyping, quality control, and imputation
The genomic DNA was isolated with the standard protocols from the peripheral blood and genotyped with the use of Infinium Asian Screening Array (Illumina, San Diego, CA, USA). This genotyping array was built using an East Asian reference panel including whole-genome sequences, designed for effectively capturing genetic variation in East Asian populations. The genotyping probe intensity was converted to SNP genotype calls using Illumina GenomeStudio version 2.0.4 (Illumina, San Diego, CA, USA). We applied stringent QC filters to the genotype data using PLINK2[29] as described previously[30]. We excluded samples with a genotyping call rate <0.98. We included only the samples of the estimated East Asian ancestry, based on the principal component analysis with the samples of HapMap project[31] (Supplementary Fig 5). We further filtered out SNPs with (i) call rate <0.99; (ii) minor allele count <5; (iii) *P*-value for Hardy-Weinberg equilibrium <1.0 × 10⁻¹⁰; and (iv) with more than 5% allele frequency difference when compared with the representative reference panels of Japanese ancestry (i.e., the reference panel used for the genotype imputation in this study and the allele frequency panel of Tohoku Medical Megabank Project[32]). After QC, we obtained genotype data of 519,668 autosomal and 17,359 X-chromosome SNPs for 1728 uRPL cases and 24,315 controls. To extend the coverage of the genetic variants to be tested, we performed genome-wide genotype imputation. We used SHAPEIT4 software[33] version 4.2.1 for haplotype phasing and Minimac4 software[34] version 1.0.1 for genotype imputation. For imputation, we used our in-house and Japanese-specific reference panel composed of *n* = 4,561 whole-genome sequence (WGS) data from multiple studies (e.g., *n* = 1939 from the BBJ study[35] and *n* = 141 WGS from the previous study[36]). Variants imputed with MAF > 0.5% and *Rsq* > 0.7 were used for the subsequent analyses.

### HLA genotype imputation
We also performed HLA genotype imputation for fine-mapping of the MHC region. We extracted the genotyped SNPs in the extended MHC

region (24-36 Mb on chromosome 6, NCBI Build 37). Based on these SNPs, we imputed the classical and non-classical HLA alleles (two-, four-, and six-digits) and corresponding amino acid sequences using DEEP*HLA[37], a multi-task convolutional deep learning method. We used the high-resolution HLA reference panel of the Japanese population[13] ($n = 1,118$). The HLA imputation procedure produced binary markers indicating the presence or absence of an investigated HLA allele or an amino acid sequence. To impute the dosage of HLA-C*12:02–HLA-B*52:01–HLA-DRB1*15:02 haplotype, we encoded each combination of the four-digit alleles of HLA-C, HLA-B, and HLA-DRB1 present in the reference panel as a single allele and trained the prediction model of DEEP*HLA. HLA variants imputed with MAF > 0.5% and an imputation quality score ($r^2$ in cross-validation) > 0.7 were used for the subsequent analyses.

### Case-control GWAS

We performed genome-wide association tests between uRPL and imputed allelic dosages using a generalized linear mixed model (GLMM) as implemented in SAIGE[11] version 0.44.6.1.26. In addition to the employment of GLMM, which controls population stratification and sample relatedness in the association test, we included the top five principal components as covariates in the regression model to robustly correct for potential population stratification. We set the genome-wide significance threshold of $P$-value < $5.0 \times 10^{-8}$.

### Sensitivity analysis accounting for MHC-associated diseases in the controls and BMI

Of the 24,315 control individuals from BBJ, 10,179 have a definitive record of disease status mapped to ICD10 codes. We confined the controls to 9,955 individuals who have BMI value and do not have a known history of diseases that potentially cause false positive association signals at MHC, including Crohn's disease (ICD10 K50), ulcerative colitis (K51), and antiphospholipid syndrome (D686). To account for the age-dependent change in BMI, we modeled BMI trajectory as a quadratic polynomial function of age. We assigned an age-matched BMI for each control individual, with age aligned to the median value in the cases. We then re-analyzed the association between the lead variant rs9263738 and uRPL with the age-matched BMI additionally incorporated into the regression model.

### Estimation of confounding biases and heritability in the uRPL GWAS

To evaluate the confounding biases and heritability in our GWAS, we performed LD score regression[38]. We used the East Asian LD score provided with the software. The liability scale heritability was calculated with the population prevalence of uRPL being 5%.

### Association analysis of the HLA variants

We performed association tests between uRPL and the imputed HLA variants using a logistic regression model as implemented in $R$ statistical software version 3.6.3. Accounting for the employment of a logistic regression model, we excluded 3rd-degree or more closely related individuals (KING[39] kingship coefficient cutoff > 0.0884) from the GWAS dataset. We assumed additive effects of the allelic dosages on a log-odds scale. We defined the HLA variants as biallelic single-nucleotide variants in the MHC region, two-, four-, and six-digits biallelic HLA alleles, biallelic HLA amino acid polymorphisms corresponding to their respective residues, and multiallelic HLA amino acid polymorphisms for each amino acid position. We incorporated the same covariates as in the GWAS sensitivity analysis into the regression model. For multiallelic amino acid variants, we estimated its significance by an omnibus test for each amino acid position by a log-likelihood ratio test, comparing the likelihood of the fitted model with the null model. The significance of the

improvement of the model fitting was evaluated by the deviance, which follows $\chi^2$ distribution with $m - 1$ degree(s) of freedom for an amino acid position with $m$ polymorphic residues. The conditional association analysis was performed to find additional HLA genes with independent uRPL risk effects by additionally including the HLA allele/haplotype as covariates in the regression model.

### Rare CNV calling

We performed haplotype-informed SNP array intensity-based CNV calling with the use of the HI-CNV[40] software version 1.0. The LRR and $\theta$ values of the genotyping probes for 625,738 autosomal variants were exported from Illumina GenomeStudio. HI-CNV was run using the phased haplotype and imputed genotype data obtained for the GWAS. The CNV calls were deduplicated and merged with the default parameters. We excluded deletions with <75 bp and duplications with > 500 bp from the subsequent analyses. We excluded 104 individuals with aberrantly high CNV calls (>50).

### Gene-based association test of CNV burden

We performed association tests between uRPL and gene-level predicted loss-of-function (pLoF) burden. Referring to canonical transcripts for 20,091 genes (from https://github.com/im3sanger/dndscv/blob/master/data/refcds_hg19.rda), if a deletion affects any part of the coding sequence or a duplication is contained within the coding sequence, we annotated the CNV as a pLoF variant[40]. We excluded 3rd-degree or more closely related individuals from the analysis. The genes of which pLoF variant carriers are more than 0.5% in cases were evaluated by using Fisher's exact two-sided test.

### Reporting summary

Further information on research design is available in the Nature Portfolio Reporting Summary linked to this article.

## Data availability

The summary statistics of the GWAS results have been deposited in the National Bioscience Database Center (NBDC) Human Database (https://humandbs.dbcls.jp/en/) under accession code hum0197 (https://humandbs.dbcls.jp/en/hum0197-latest). Data can also be browsed at our pheweb.jp[41] website (https://pheweb.jp/). GWAS genotype data of the BBJ are available at the NBDC Human Database (research ID: hum0311).

## Code availability

We used publicly available software for the data analysis. The software used is described in the Methods section.

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

## Acknowledgements

The authors sincerely thank all the participants involved in this study. This research was supported by the KAKENHI Grants-in-Aid from the Japanese Society for the Promotion of Science (JSPS) [grant number 23K14451 (to K.S.), 23K08850 (to Y.Y.), and 22H00476 (to Y.O.)], the Japan Agency for Medical Research and Development (AMED) [grant number JP23km0405211, JP23km0405217, JP23ek0109594, JP23ek0410113, JP23kk0305022, JP223fa627002, JP223fa627010, JP233fa627011, JP23zf0127008, JP23tm0524002 (to Y.O.)], JST Moonshot R&D [grant number JPMJMS2021 and JPMJMS2024 (to Y.O.)], the Japanese Ministry of Education, Science, and Technology (MEXT) Promotion of Distinctive Joint Research Center Program [grant number JPMXP0621467963 (to M.S.-O.)], the Takeda Science Foundation, Bioinformatics Initiative of Osaka University Graduate School of Medicine, Institute for Open and Transdisciplinary Research Initiatives, Center for Infectious Disease Education and Research (CiDER), and Center for Advanced Modality and DDS (CAMaD), Osaka University.

## Author contributions

K.S., Y.Y., Y.O., and M.S.-O. designed the study and wrote the manuscript. K.S., Y.Y., T. Naito, T.O., T. Nishiyama, Y.O., and M.S.-O. conducted data analysis. S.G., H.Y., F.O., and T.K. collected the samples. K.M. and the members of the Biobank Japan Project constructed the data. Y.O., and M.S.-O. supervised the study.

## Competing interests

The authors declare no competing interests.

## Additional information

[1]Department of Genome Informatics, Graduate School of Medicine, the University of Tokyo, Tokyo, Japan. [2]Department of Statistical Genetics, Osaka University Graduate School of Medicine, Osaka, Suita, Japan. [3]Laboratory for Systems Genetics, RIKEN Center for Integrative Medical Sciences, Yokohama, Japan. [4]Department of Obstetrics and Gynecology, Nagoya City University Graduate School of Medical Sciences, Nagoya, Japan. [5]Department of Public Health, Nagoya City University Graduate School of Medical Sciences, Nagoya, Japan. [6]Laboratory of Genome Technology, Human Genome Center, Institute of Medical Science, The University of Tokyo, Tokyo, Japan. [7]Laboratory of Clinical Genome Sequencing, Department of Computational Biology and Medical Sciences, Graduate School of Frontier Sciences, the University of Tokyo, Tokyo, Japan. [8]Laboratory of Statistical Immunology, Immunology Frontier Research Center (WPI-IFReC), Osaka University, Osaka, Suita, Japan. [9]Premium Research Institute for Human Metaverse Medicine (WPI-PRIMe), Osaka University, Osaka, Suita, Japan. [33]These authors contributed equally: Kyuto Sonehara, Yoshitaka Yano. [34]These authors jointly supervised this work: Yukinori Okada, Mayumi Sugiura-Ogasawara. ✉e-mail: yuki-okada@m.u-tokyo.ac.jp; og.mym@med.nagoya-cu.ac.jp

## the Biobank Japan Project

Koichi Matsuda ⓘ [6,7], Yuji Yamanashi[10], Yoichi Furukawa[11], Takayuki Morisaki[12], Yukinori Okada ⓘ [1,2,3,8,9,34]✉, Yoshinori Murakami[13], Yoichiro Kamatani[7,14], Kaori Muto[15], Akiko Nagai[7], Yusuke Nakamura[16], Wataru Obara[17], Ken Yamaji[18], Kazuhisa Takahashi[19], Satoshi Asai[20,21], Yasuo Takahashi[21], Shinichi Higashiue[22], Shuzo Kobayashi[22], Hiroki Yamaguchi[23], Yasunobu Nagata[23], Satoshi Wakita[23], Chikako Nito[24], Yu-ki Iwasaki[25], Shigeo Murayama[26], Kozo Yoshimori[27], Yoshio Miki[28], Daisuke Obata[29], Masahiko Higashiyama[30], Akihide Masumoto[31], Yoshinobu Koga[31] & Yukihiro Koretsune[32]

[10]Division of Genetics, The Institute of Medical Science, The University of Tokyo, Tokyo, Japan. [11]Division of Clinical Genome Research, Institute of Medical Science, The University of Tokyo, Tokyo, Japan. [12]Department of Computational Biology and Medical Sciences, Graduate School of Frontier Sciences, BioBank Japan, Institute of Medical Science, The University of Tokyo, Tokyo, Japan. [13]Department of Cancer Biology, Institute of Medical Science, The University of Tokyo, Tokyo, Japan. [14]Laboratory of Complex Trait Genomics, Graduate School of Frontier Sciences, The University of Tokyo, Tokyo, Japan. [15]Department of Public Policy, Institute of Medical Science, The University of Tokyo, Tokyo, Japan. [16]The Institute of Medical Science, The University of Tokyo, Tokyo, Japan. [17]Department of Urology, Iwate Medical University, Iwate, Japan. [18]Department of Internal Medicine and Rheumatology, Juntendo University Graduate School of Medicine, Tokyo, Japan. [19]Department of Respiratory Medicine, Juntendo University Graduate School of Medicine, Tokyo, Japan. [20]Division of Pharmacology, Department of Biomedical Science, Nihon University School of Medicine, Tokyo, Japan. [21]Division of Genomic Epidemiology and Clinical Trials, Trials Research Center, Nihon University. School of Medicine, Tokyo, Japan. [22]Tokushukai Group, Tokyo, Japan. [23]Department of Hematology, Nippon Medical School, Tokyo, Japan. [24]Laboratory for Clinical Research, Collaborative Research Center, Nippon Medical School, Tokyo, Japan. [25]Department of Cardiovascular Medicine, Nippon Medical School, Tokyo, Japan. [26]Tokyo Metropolitan Geriatric Hospital and Institute of Gerontology, Tokyo, Japan. [27]Fukujuji Hospital, Japan Anti-Tuberculosis Association, Tokyo, Japan. [28]Center for Clinical Research and Advanced Medicine, Shiga University of Medical Science, Shiga, Japan. [29]The Cancer Institute Hospital of the Japanese Foundation for Cancer Research, Tokyo, Japan. [30]Department of General Thoracic Surgery, Osaka International Cancer Institute, Osaka, Japan. [31]IIZUKA HOSPITAL, Fukuoka, Japan. [32]National Hospital Organization Osaka National Hospital, Osaka, Japan.

