## [Peer Review File · Nature Communications]

Common and rare genetic variants predisposing females to unexplained recurrent pregnancy lossREVIEWER COMMENTS

Reviewer #1 (Remarks to the Author):

Sonehara et al. identifies the genetic basis of unexplained recurrent pregnancy loss (uRPL). They perform GWAS and genome-wide copy number variation (CNV) analysis in a cohort of Japanese ancestry. They identify HLA alleles with a protective effect and CNV at the cadherin-11 gene that increase risk for RPL.

Overall, this work is an important contribution to human genetics and reproductive immunology. It has several strengths. Recurrent pregnancy loss is a heterogenous phenotype with a limited number of therapeutic interventions. GWAS studies using large-scale biobanks have not robustly investigated reproductive traits, especially pregnancy loss. This work builds on a growing body of evidence pointing to the importance of immunologic mechanisms in reproductive biology.

The careful phenotyping of cases to enrich for genetic etiologies of uRPL is another strength of this study; for example, the stratified analysis systematically excluding common comorbid diseases associated with pregnancy loss. Another major contribution to the field of Human Genetics is the use of a non-European population and identifying a population specific locus associated with uRPL. Finally, while the genetics of pregnancy loss has been investigated before, very few studies have used a genome-wide approach, and I cannot find another study with a similar magnitude of sample size and in this specific population.

Key technical aspects of a genome-wide study appear to be conducted appropriately. Cases and controls are drawn from people of Japanese descent. Sample relatedness and shared ancestry is controlled for using a GLMM with 5 PCs as covariates. The QQ plot (Supp Fig 1) qualitatively doesn't reveal any systemic inflation of p-values with appropriate λ_{GC} and LD score regression intercept. A conventional p-value accounting for multiple testing is used and Manhattan plots demonstrate expected patterns. Conditional analyses on HLA-C/B/DRB1 (Figure 2) provides compelling evidence for the long-range Japanese specific haplotype.

While the stratified analysis is interesting, I worry about over interpretation of these results. This is because as the authors performed further phenotypic refinement, the sample size for the cases decreased dramatically. Thus, interpreting null results raises questions about the statistical power of such analyses. However, I appreciate the authors including the analyses.

I provide below a few major and minor comments in hopes of strengthening this manuscript.

Major Points

- Please provide further details on how cases and controls were defined.
 - o Lines 312-314: For the 1,800 women enrolled with two or more unexplained pregnancy loss: how was this ascertained? Was it through self-report, interview, questionnaire, or through a doctor?
 - o Line 320-321: Were any exclusion criteria applied to the selection of control women from the Biobank Japan Project?
 - o While Table 1 provides a great description of the controls, can the authors also include such descriptions for the controls? I am most concerned about difference in age (doi: <https://doi.org/10.1136/bmj.l869>) and body mass index (doi: [10.4103/2230-8229.102316](https://doi.org/10.4103/2230-8229.102316), <https://doi.org/10.1038/s41598-021-86445-2>). Finally, what was the prevalence of pregnancy loss in the Biobank Japan Project?
 - o Since the controls are not as stringently ascertained, I worry about differences in other diseases leading to the HLA signal in their GWAS. For example, if the control samples had a very high

prevalence for ulcerative colitis and had a variant with high effect size, theoretically, the GWAS could detect a false positive signal. I believe this to be a more theoretical risk as the number of control samples are about x13 larger than the cases. One or two lines making this limitation explicit in the discussion should be included.

o I realize that addressing age and BMI maybe technically challenging. One immediate idea is to generate age-matched BMI value from the controls, where the age would be age at sampling for each case. If this is deemed too technically challenging, this limitation must be addressed in the discussion.

- Line 199-201: The statement "If so, restricting the GWAS..." requires further clarification. If after restricting the cases to ANA+, the effect size increased, then it is fair to suggest that ANA+ may, through the HLA locus, influence uRPL. However, the lack of an increase in effect size is difficult to interpret. The mechanism of ANA+ mediated uRPL may act through a different genetic locus. I would recommend modifying the wording in this sentence for better clarity.

- Line 334-334. "We included only samples of the estimated East Asian ancestry, based on PCA with the samples of the HapMap Project." Please provide more details on how samples were ascertained based on ancestry.

Minor Points

- In Table 2, the risk allele is noted to be "C". Line 138 refers to the "T" allele as the risk allele. Can you clarify?

Reviewer #2 (Remarks to the Author):

The manuscript "Common and rare genetic variants predisposing to unexplained recurrent pregnancy loss" by Sonehara et al aims to perform a GWAS on 1728 unexplained recurrent pregnancy loss cases and 24,353 controls of Japanese ancestry. The authors report a signal in the MHC region and additionally find an association with a rare loss-of-function CNV of the cadherin-11 gene. As such, it is the largest genetic study of recurrent pregnancy loss and potentially valuable to the field.

Overall, the methodology is standard in the GWAS field and necessary quality control and multiple testing corrections have been applied.

However, I have found a few issues that need clarifying to be sure the presented results are robust and support the conclusions.

First, it is unclear who were the controls - a random selection of women from Biobank Japan or only women who had been pregnant/delivered? Were the same exclusion criteria used in selecting cases also applied in controls? Using exclusions only in cases and not in controls may introduce bias, which I think is especially relevant in this case, considering that APS cases (which likely involve the MHC region in its etiopathogenesis) were excluded from the cases. Please also include basic characteristics (age, BMI, parity) for controls in Table 1.

Second, on page 15 it is stated that "By assuming that uRPL with different embryonic karyotypes involves different etiologies, we excluded the cases showing embryonic karyotype abnormalities, defining the remaining 1,496 cases as case group". However, in Table 1 it can be seen that the embryonic karyotype was unknown for 1,292 women, therefore including them in a group without karyotype abnormalities is not entirely correct.

Third, would it be possible to validate the CDH11 CNV with a different method, either with a different bioinformatic tool or with qPCR?

Minor comments:

-Please clarify if the karyotypes of the abortuses refer to the latest miscarriage? How many women had had miscarriages with multiple partners?

- Please also provide the CI-s and effect allele frequencies in the abstract.

Response to the reviewers' comments:

Reviewer #1 (Remarks to the Author):

Sonehara et al. identifies the genetic basis of unexplained recurrent pregnancy loss (uRPL). They perform GWAS and genome-wide copy number variation (CNV) analysis in a cohort of Japanese ancestry. They identify HLA alleles with a protective effect and CNV at the cadherin-11 gene that increase risk for RPL.

Overall, this work is an important contribution to human genetics and reproductive immunology. It has several strengths. Recurrent pregnancy loss is a heterogeneous phenotype with a limited number of therapeutic interventions. GWAS studies using large-scale biobanks have not robustly investigated reproductive traits, especially pregnancy loss. This work builds on a growing body of evidence pointing to the importance of immunologic mechanisms in reproductive biology.

The careful phenotyping of cases to enrich for genetic etiologies of uRPL is another strength of this study; for example, the stratified analysis systematically excluding common comorbid diseases associated with pregnancy loss. Another major contribution to the field of Human Genetics is the use of a non-European population and identifying a population specific locus associated with uRPL. Finally, while the genetics of pregnancy loss has been investigated before, very few studies have used a genome-wide approach, and I cannot find another study with a similar magnitude of sample size and in this specific population.

Key technical aspects of a genome-wide study appear to be conducted appropriately. Cases and controls are drawn from people of Japanese descent. Sample relatedness and shared ancestry is controlled for using a GLMM with 5 PCs as covariates. The QQ plot (Supp Fig 1) qualitatively doesn't reveal any systemic inflation of p-values with appropriate Λ_{GC} and LD score regression intercept. A conventional p-value accounting for multiple testing is used and Manhattan plots demonstrate expected patterns. Conditional analyses on HLA-C/B/DRB1 (Figure 2) provides compelling evidence for the long-range Japanese specific haplotype.

While the stratified analysis is interesting, I worry about over interpretation of these results. This is because as the authors performed further phenotypic refinement, the sample size for the cases decreased dramatically. Thus, interpreting null results raises questions about the statistical power of such analyses. However, I appreciate the authors including the analyses.

I provide below a few major and minor comments in hopes of strengthening this manuscript.

Response:

We sincerely thank you for the careful assessment of our manuscript and for acknowledging the contribution of our study to the field. We also appreciate the points raised by you, which we have addressed in this revision, and we think the current manuscript is substantially improved. Please find our point-by-point responses below.

Major Points

· Please provide further details on how cases and controls were defined.

Lines 312-314: For the 1,800 women enrolled with two or more unexplained pregnancy loss: how was this ascertained? Was it through self-report, interview, questionnaire, or through a doctor?

Response:

Thank you for raising an important point. The 1,800 women were patients who visited the Nagoya City University Hospital for a consultation for pregnancy loss. Each patient's history of pregnancy loss was ascertained through medical interviews by the obstetricians.

Modification:

To clarify the case definition, we added the description as;

"We enrolled 1,800 Japanese patients with a history of two or more unexplained pregnancy losses. All patients were recruited from Nagoya City University Hospital between May 2007 and July 2022. All medical information, including the history of pregnancy losses, was obtained through medical interviews by the obstetricians. Chemical pregnancies were not included in pregnancy losses. All patients underwent a systematic examination, including 3D-ultrasound sonography, chromosome analysis of both partners, determination of antiphospholipid antibody, including lupus anticoagulant, by diluted activated partial prothrombin time, diluted Russell viper venom time and β 2 glycoprotein I-dependent anticardiolipin antibody, and blood tests for hypothyroidism and diabetes mellitus, before a subsequent pregnancy." in **Methods** (page 25, lines 360–369).

Line 320-321: Were any exclusion criteria applied to the selection of control women from the Biobank Japan Project?

Response:

Thank you for raising an important point. In this study, we included all available female participants of East Asian ancestry in the Biobank Japan Project (BBJ) with the aim of maximizing statistical power for detecting susceptibility loci. Nonetheless, we agree with you that our analytical approach could potentially lead to some readers' concerns, as you pointed out in the comments provided later. We addressed this issue by adding sensitivity analyses accounting for the body mass index as well as the prevalence of immune-related diseases in the controls in response to your comments below.

Modification:

We revised the manuscript to incorporate the additionally conducted sensitivity analyses in response to the reviewer's comment. The details of the modification are described in the following responses.

While Table 1 provides a great description of the controls, can the authors also include such descriptions for the controls? I am most concerned about difference in age (doi: <https://doi.org/10.1136/bmj.l869> <<https://doi.org/10.1136/bmj.l869>>) and body mass index (doi: [10.4103/2230-8229.102316](https://doi.org/10.4103/2230-8229.102316), <https://doi.org/10.1038/s41598-021-86445-2> <<https://doi.org/10.1038/s41598-021-86445-2>>). Finally, what was the prevalence of pregnancy loss in the Biobank Japan Project?

Response:

Thank you for the insightful advice. We added the descriptions of age and body mass index to **Table 1** according to your suggestion. While checking the clinical information of the control individuals, we found some individuals whose reported sex was inconsistent with the genotype data and excluded them from the analysis.

While we agree with you that the prevalence of pregnancy loss in the controls is an important point, we would like to clarify the aim of our study design. Since the target diseases of BBJ do not include pregnancy loss, not all the participants were examined for the medical history of pregnancy loss. To put it the other way around, the controls were neither enriched nor depleted for the patients with pregnancy loss, and the prevalence of uRPL in the controls is expected to be similar to that in the general Japanese population (*i.e.*, ~2%). By comparing the ascertained uRPL cases with this general population, we can evaluate the enrichment of the susceptibility alleles.

Importantly, while we admit that some control participants may have uRPL, given the direction of the effect, it will not lead to false positive associations but conservative results. The potential impureness of the controls rather suggests that the association between the MHC locus and uRPL could be underestimated in our study. In other words, the odds ratio can be estimated higher when the association is tested using a pure control cohort, suggesting the importance of MHC in the genetic predisposition to uRPL. We clarified these points in **Discussion**.

Modification:

According to the reviewer’s suggestion, we added the descriptions of age and body mass index to **Table 1**.

(revised) Table 1. Characteristics of study participants (n = 1,728 cases and 24,315 controls)†

Characteristics	Case		Control	
	n	(%)	n	(%)
Age at sampling				
<20	1	0.0	115	0.5
20-29	329	19.0	335	1.4
30-39	1158	67.0	1488	6.1
≥40	240	13.9	22,377	92.0
Body Mass Index				
<18.5	272	15.7	3405	14.0
18.5-24.9	1275	73.8	15,882	65.3
≥25	163	9.4	4471	18.4

Missing	18	1.0				
Number of previous pregnancy loss			early miscarriage	intrauterine fetal death	chrioamnionitis	
0	0	0.0	0	0	0	
2	884 [‡]	51.1	864	71	25	
3	586	33.9	580	62	22	
≥4	257	14.9	257	38	17	
Number of previous live births						
0	1330	77.0				
≥1	398	23.0				
Embryonic (fetal) karyotype						
euploid	204	11.8				
aneuploid	125	7.2				
45,X	46	2.7				
triploid	31	2.4				
others	30	1.7				
unknown	1292	74.8				
Antinuclear antibody						
negative	1092	63.2				
positive (≥1:40)	543	31.4				
missing	93	5.4				
Hypothyroidism						
presence free T4 ≤0.9	285	16.5				
absence	1336	77.3				
missing	107	6.2				
Autoimmune disease						
systemic lupus erythematosus	1	0.1				
rheumatoid arthritis	5	0.3				
chronic thyroiditis	98	5.7				
absence	1584	91.7				
others	40	2.3				

†Including 65 cases diagnosed as weakly positive for antiphospholipid antibody

‡Including 2 cases in which the patient was diagnosed with a history of two miscarriages, but one of the two miscarriages was later diagnosed as a chemical pregnancy.

To clarify the influence of the potential impureness of the controls on the GWAS, we added the description as;

“Our study has some potential limitations. First, the control participants for the association analyses were derived from the existing biobank, and not all of them were confirmed to be free of uRPL. However, the potential misclassification of the controls in our GWAS does not undermine the robustness of our findings. Even if some controls had a history of pregnancy loss, it would not lead to false positive associations but rather conservative results with underestimated heritability and odds ratio.” in **Discussion** (page 24, lines 336–341).

Since the controls are not as stringently ascertained, I worry about differences in other diseases leading to the HLA signal in their GWAS. For example, if the control samples had a very high prevalence for ulcerative colitis and had a variant with high effect size, theoretically, the GWAS could detect a false positive signal. I believe this to be a more theoretical risk as

the number of control samples are about x13 larger than the cases. One or two lines making this limitation explicit in the discussion should be included.

Response:

Thank you for the thoughtful comment. Although ulcerative colitis (UC) is not included in the target diseases of BBJ, we admit that the hospital-based cohort design of BBJ could potentially result in a high prevalence of UC in the study participants. Among the control individuals, 10,179 were recently follow-up surveyed for the disease status mapped to the ICD10 codes. Referring to this survey data, we restricted the controls to 9,955 individuals without a known history of diseases that potentially cause false positive association signals, including Crohn's disease (ICD10 K50), UC (K51), and antiphospholipid syndrome (D686). When we repeated a case–control association analysis using this stringently defined control set as a sensitivity analysis (with adjustment for the age-matched BMI in response to the next comment by you), the lead SNP rs9263738 remained genome-wide significant ($P = 7.5 \times 10^{-10}$) with an odds ratio comparable to that before the case restriction (OR = 1.52 [95%CI 1.33–1.74] after the case restriction; OR = 1.51 [95%CI 1.33–1.72] before the case restriction). Nonetheless, given the wide range of human complex traits associated with HLA variants, we admit that the association signal could be affected by the disease prevalence in the controls. We included these points in **Discussion**.

Modification:

To incorporate the sensitivity analysis result into the manuscript, we added the description as;

“In our primary GWAS above, the control group included all the female participants available in the Biobank Japan project (BBJ) to maximize statistical power for detecting susceptibility loci. However, given the hospital-based cohort design of BBJ, our primary GWAS could have the following potential limitations which may introduce false positive associations: i) high prevalence for other MHC-associated diseases in the controls and ii) difference in the body mass index (BMI), a known risk factor for RPL, between cases and controls. To address these potential limitations, we additionally performed a sensitivity analysis. We confined the controls to 9,955 individuals who had a definitive record of the ICD10 code and were explicitly free of Crohn's disease (ICD10 K50), ulcerative colitis (K51), and antiphospholipid syndrome (D686). To account for the BMI difference, we estimated age-matched BMI based on the relationship between age and BMI (Methods). When we repeated the association test using the strictly defined control set with adjustment for the age-matched BMI, the lead variant rs9263738 remained genome-wide significant with comparable effect size ($P = 7.5 \times 10^{-10}$; OR = 1.52 [95% CI = 1.33–1.74]), indicating the robustness of the association between uRPL and MHC.” in **Results** (page 7–8, lines 144–158).

We also added a new subsection as;

“Sensitivity analysis accounting for MHC-associated diseases in the controls and BMI

Of the 24,315 control individuals from BBJ, 10,179 have a definitive record of disease status mapped to ICD10 codes. We confined the controls to 9,955 individuals who have BMI value and do not have a known history of diseases that potentially cause false positive association signals at MHC, including Crohn's disease (ICD10 K50), ulcerative colitis (K51), and antiphospholipid syndrome (D686). To account for the age-dependent change in BMI, we modeled BMI trajectory as a quadratic polynomial function of age. We assigned an age-matched BMI for each control individual, with age aligned to the median value in the cases. We then re-analyzed the association between the lead variant rs9263738 and uRPL with the

age-matched BMI additionally incorporated into the regression model.” in **Methods** (page 27–28, lines 429–438).

To clarify the potential limitation, we added the description as;

“Our study has some potential limitations. ... Second, although we performed the sensitivity analysis to account for the patients of MHC-associated diseases in the controls, the highly pleiotropic nature of the MHC locus could affect the association signal by other MHC-associated diseases potentially enriched in the controls.” in **Discussion** (page 24, lines 336–344).

I realize that addressing age and BMI maybe technically challenging. One immediate idea is to generate age-matched BMI value from the controls, where the age would be age at sampling for each case. If this is deemed too technically challenging, this limitation must be addressed in the discussion.

Response:

Thank you for raising an important point. Following your suggestion, we modeled the age-BMI relationship as a polynomial function based on the control data (**Response Figure 1**). According to this function, we generated age-matched BMI for the control participants with the age being the median value in the cases (i.e., 34.1). We repeated an association analysis with the age-matched BMI incorporated into the regression model and confirmed that the lead SNP rs9263738 remained genome-wide significant ($P = 7.5 \times 10^{-10}$; OR = 1.52 [95%CI 1.33–1.74]). Nonetheless, we admit that the age-matched BMI estimation above described is a relatively simple approach and may not fully adjust for the BMI difference at reproductive age between cases and controls. We clarified these points in **Discussion**.

Response Figure 1 | age-BMI relationship in the BBJ controls

Relationship between body mass index (BMI) and age is shown. Black markers represent BBJ control individuals. The non-linear relationship between age and BMI was modeled as a quadratic polynomial function, which is indicated as the blue curved line.

Modification:

To incorporate the sensitivity analysis result into the manuscript, we added the description as;

“In our primary GWAS above, the control group included all the female participants available in the Biobank Japan project (BBJ) to maximize statistical power for detecting susceptibility loci. However, given the hospital-based cohort design of BBJ, our primary GWAS could have the following potential limitations which may introduce false positive associations: i) high prevalence for other MHC-associated diseases in the controls and ii) difference in the body mass index (BMI), a known risk factor for RPL, between cases and controls. To address these potential limitations, we additionally performed a sensitivity analysis. We confined the controls to 9,955 individuals who had a definitive record of the ICD10 code and were explicitly free of Crohn’s disease (ICD10 K50), ulcerative colitis (K51), and antiphospholipid syndrome (D686). To account for the BMI difference, we estimated age-matched BMI based on the relationship between age and BMI (Methods). When we repeated the association test using the strictly defined control set with adjustment for the age-matched BMI, the lead variant rs9263738 remained genome-wide significant with comparable effect size ($P = 7.5 \times 10^{-10}$; OR = 1.52 [95% CI = 1.33–1.74]), indicating the robustness of the association between uRPL and MHC.” in **Results** (page 7–8, lines 144–158).

We also added a new subsection as;

“Sensitivity analysis accounting for MHC-associated diseases in the controls and BMI

Of the 24,315 control individuals from BBJ, 10,179 have a definitive record of disease status mapped to ICD10 codes. We confined the controls to 9,955 individuals who have BMI value and do not have a known history of diseases that potentially cause false positive association signals at MHC, including Crohn’s disease (ICD10 K50), ulcerative colitis (K51), and antiphospholipid syndrome (D686). To account for the age-dependent change in BMI, we modeled BMI trajectory as a quadratic polynomial function of age. We assigned an age-matched BMI for each control individual, with age aligned to the median value in the cases. We then re-analyzed the association between the lead variant rs9263738 and uRPL with the age-matched BMI additionally incorporated into the regression model.” in **Methods** (page 27–28, lines 429–438).

To clarify the potential limitation, we added the description as;

“Our study has some potential limitations. ... Third, in the sensitivity analysis, we adjusted for BMI by modeling the BMI trajectory as a function of age. Our model was relatively simple and may not fully capture the potential difference in the BMI difference at reproductive age between cases and controls.” in **Discussion** (page 24, lines 336–347).

Line 199-201: The statement “If so, restricting the GWAS...” requires further clarification. If after restricting the cases to ANA+, the effect size increased, then it is fair to suggest that ANA+ may, through the HLA locus, influence uRPL. However, the lack of an increase in effect size is difficult to interpret. The mechanism of ANA+ mediated uRPL may act through a different genetic locus. I would recommend modifying the wording in this sentence for better clarity.

Response and modification:

Thank you for the thoughtful comment. We fully agree with you that if the HLA locus is not involved in the mechanism of ANA-mediated uRPL, changes in effect size would not be observed even if ANA mediates uRPL pathogenesis.

To clarify the implicit assumption behind our analysis and the caveat in interpreting the derived results, we revised the descriptions as;

“Given that the MHC region is involved in a wide range of autoimmune diseases susceptibility, we hypothesized that the association between MHC and uRPL may be mediated by ANA production. If so, restricting the GWAS cases to those with positive ANA would result in a substantially larger effect size of the lead HLA haplotype HLA-C*12:02–HLA-B*52:01–HLA-DRB1*15:02. Despite this expectation, the odds ratio was nearly equivalent whether limiting to positive ANA or not (Fig. 3). We note that this observation does not refute ANA involvement in the etiology of uRPL, since ANA may mediate uRPL pathogenesis independent of the MHC locus.” in **Results** (page 17, lines 221–228).

Line 334-334. “We included only samples of the estimated East Asian ancestry, based on PCA with the samples of the HapMap Project.’ Please provide more details on how samples were ascertained based on ancestry.

Response and modification:

Thank you for the thoughtful suggestion. To show more details on the ancestry ascertainment of the study participants in our GWAS, we added **Supplementary Fig. 5** below.

Supplementary Figure 5 | Visualization of the principal component vectors of the GWAS participants

The distributions of principal components of the genotypes of the study participants are indicated together with the individuals of the HapMap project. Each marker represents an individual. CHB, Han Chinese in Beijing, China; JPT, Japanese in Tokyo, Japan; CEU, Utah residents with Northern and Western European ancestry; YRI, Yoruba in Ibadan, Nigeria.

Minor Points

In Table 2, the risk allele is noted to be “C”. Line 138 refers to the “T” allele as the risk allele. Can you clarify?

Response and modification:

Thank you for pointing it out. “T” allele is the risk allele. We revised **Table 2** accordingly.

(revised) Table 2. Association summary of the lead genetic variant

rsID	Chr	Position (hg19)	Alleles	Risk Allele	Risk Allele Frequency		OR (95% CI)	P-value
					Case	Control		
rs9263738	6	31,109,767	T/C	T	0.893	0.871	1.51 (1.33–1.72)	1.4 × 10 ⁻¹⁰

Reviewer #2 (Remarks to the Author):

The manuscript "Common and rare genetic variants predisposing to unexplained recurrent pregnancy loss" by Sonehara et al aims to perform a GWAS on 1728 unexplained recurrent pregnancy loss cases and 24,353 controls of Japanese ancestry. The authors report a signal in the MHC region and additionally find an association with a rare loss-of-function CNV of the cadherin-11 gene. As such, it is the largest genetic study of recurrent pregnancy loss and potentially valuable to the field.

Overall, the methodology is standard in the GWAS field and necessary quality control and multiple testing corrections have been applied.

However, I have found a few issues that need clarifying to be sure the presented results are robust and support the conclusions.

Response:

We truly appreciate you for acknowledging the value and methodology of the GWAS in our work. We also thank you for your important comments. We have fully addressed your comments and believe our manuscript should be significantly improved. Please find our point-by-point responses below.

First, it is unclear who were the controls - a random selection of women from Biobank Japan or only women who had been pregnant/delivered? Were the same exclusion criteria used in selecting cases also applied in controls? Using exclusions only in cases and not in controls may introduce bias, which I think is especially relevant in this case, considering that APS cases (which likely involve the MHC region in its etiopathogenesis) were excluded from the cases. Please also include basic characteristics (age, BMI, parity) for controls in Table 1.

Response:

Thank you for raising an important point. Among the control individuals, 10,179 were recently follow-up surveyed for the ICD10 codes. Referring to the survey results, we restricted the controls to 9,955 individuals who don't have a known history of diseases that potentially cause false positive association signals, including antiphospholipid syndrome (ICD10 D686). When we repeated a case-control association analysis using this stringently defined control set, the lead SNP rs9263738 remained genome-wide significant ($P = 3.1 \times 10^{-10}$; OR = 1.53 [95%CI 1.34–1.76]).

We also added the descriptions of age and body mass index to **Table 1** according to your suggestion. Regarding parity, while we agree with you that the prevalence of pregnancy loss in the controls is an important point, we would like to clarify the aim of our study design. Since the target diseases of Biobank Japan do not include pregnancy loss, not all the participants were examined for the medical history of pregnancy loss. To put it the other way around, the controls were neither enriched nor depleted for the patients with pregnancy loss, and the prevalence in the controls is expected to be similar to that in the general Japanese population (i.e., ~2%). By comparing the ascertained uRPL cases with this general population, we can evaluate the enrichment of the susceptibility alleles.

Importantly, while we admit that some control participants may have uRPL, given the direction of the effect, it will not lead to false positive associations but conservative results. The potential impureness of the controls rather suggests that the association between the MHC locus and uRPL could be underestimated in our study. In other words, the odds ratio can be estimated higher when the association is tested using a pure control cohort, suggesting the importance of MHC in the genetic predisposition to uRPL. We clarified these points in **Discussion**.

Modification:

To incorporate the sensitivity analysis result into the manuscript, we added the description as;

“In our primary GWAS above, the control group included all the female participants available in the Biobank Japan project (BBJ) to maximize statistical power for detecting susceptibility loci. However, given the hospital-based cohort design of BBJ, our primary GWAS could have the following potential limitations which may introduce false positive associations: i) high prevalence for other MHC-associated diseases in the controls and ii) difference in the body mass index (BMI), a known risk factor for RPL, between cases and controls. To address these potential limitations, we additionally performed a sensitivity analysis. We confined the controls to 9,955 individuals who had a definitive record of the ICD10 code and were explicitly free of Crohn’s disease (ICD10 K50), ulcerative colitis (K51), and antiphospholipid syndrome (D686). To account for the BMI difference, we estimated age-matched BMI based on the relationship between age and BMI (Methods). When we repeated the association test using the strictly defined control set with adjustment for the age-matched BMI, the lead variant rs9263738 remained genome-wide significant with comparable effect size ($P = 7.5 \times 10^{-10}$; OR = 1.52 [95% CI = 1.33–1.74]), indicating the robustness of the association between uRPL and MHC.” in **Results** (page 7–8, lines 144–158).

We also added a new subsection as;

Sensitivity analysis accounting for MHC-associated diseases in the controls and BMI

Of the 24,315 control individuals from BBJ, 10,179 have a definitive record of disease status mapped to ICD10 codes. We confined the controls to 9,955 individuals who have BMI value and do not have a known history of diseases that potentially cause false positive association signals at MHC, including Crohn’s disease (ICD10 K50), ulcerative colitis (K51), and antiphospholipid syndrome (D686). To account for the age-dependent change in BMI, we modeled BMI trajectory as a quadratic polynomial function of age. We assigned an age-matched BMI for each control individual, with age aligned to the median value in the cases. We then re-analyzed the association between the lead variant rs9263738 and uRPL with the age-matched BMI additionally incorporated into the regression model.” in **Methods** (page 27–28, lines 429–438).

According to the reviewer’s suggestion, we added the descriptions of age and body mass index to **Table 1**.

(revised) Table 1. Characteristics of study participants (n = 1728 cases and 24,315 controls)†

Characteristics	Case		Control	
	n	(%)	n	(%)
Age at sampling				
<20	1	0.0	115	0.5

20-29	329	19.0				335	1.4
30-39	1158	67.0				1488	6.1
≥40	240	13.9				22,377	92.0
Body Mass Index							
<18.5	272	15.7				3405	14.0
18.5-24.9	1275	73.8				15,882	65.3
≥25	163	9.4				4471	18.4
Missing	18	1.0				557	2.3
Number of previous pregnancy loss							
			early miscarriage	intrauterine fetal death	chrioamnionitis		
0	0	0.0	0	0	0		
2	884 [†]	51.1	864	71	25		
3	586	33.9	580	62	22		
≥4	257	14.9	257	38	17		
Number of previous live births							
0	1330	77.0					
≥1	398	23.0					
Embryonic (fetal) karyotype							
euploid	204	11.8					
aneuploid	125	7.2					
45,X	46	2.7					
triploid	31	2.4					
others	30	1.7					
unknown	1292	74.8					
Antinuclear antibody							
negative	1092	63.2					
positive (≥1:40)	543	31.4					
missing	93	5.4					
Hypothyroidism							
presence free T4 ≤0.9	285	16.5					
absence	1336	77.3					
missing	107	6.2					
Autoimmune disease							
systemic lupus erythematosus	1	0.1					
rheumatoid arthritis	5	0.3					
chronic thyroiditis	98	5.7					
absence	1584	91.7					
others	40	2.3					

†Including 65 cases diagnosed as weakly positive for antiphospholipid antibody

‡Including 2 cases in which the patient was diagnosed with a history of two miscarriages, but one of the two miscarriages was later diagnosed as a chemical pregnancy.

Second, on page 15 it is stated that "By assuming that uRPL with different embryonic karyotypes involves different etiologies, we excluded the cases showing embryonic karyotype abnormalities, defining the remaining 1,496 cases as case group". However, in Table 1 it can be seen that the embryonic karyotype was unknown for 1,292 women, therefore including them in a group without karyotype abnormalities is not entirely correct.

Response:

Thank you for the thoughtful comment. We agree with you that our wording for referring to the stratified case groups in the previous manuscript could be misleading. We would like first to clarify our purpose for excluding cases with ascertained aneuploidy in the stratified analysis. Given the large-scale alterations in the genome due to aneuploidy, we assumed that aneuploidy would in itself be a predominant cause of miscarriage relative to the common HLA polymorphism focused on our analysis. We aim to reduce the heterogeneity in the underlying biology behind the uRPL cases to make clear differences in the HLA allele's effect size across the immunological strata (*i.e.*, ANA-positive and autoimmunity-negative).

While we admit that it would be ideal to perform our stratified analysis with the cases restricted to those who were with ascertained euploidy, if doing so the case sample size was too limited to provide insights. Considering the trade-off between the purity in the etiological background and the sample size, we performed the stratified analysis with the cases with confirmed embryonic euploid or undetermined karyotype. We would like to ask you to kindly note that we did not propose any euploid miscarriage-specific association by the stratified analyses.

Modification:

To clarify our purpose in the stratified analysis, we revised the descriptions as;

“Given the heterogeneity in the underlying biology of uRPL, we stratified the GWAS participants according to clinical features, such as the embryonic karyotype and immunological characteristics. We assumed that uRPL with different embryonic karyotypes involves different etiologies; in particular, abnormal embryonic karyotypes would in itself serve as a predominant cause of miscarriage. To reduce heterogeneity in the etiological background of the uRPL cases, we excluded the cases showing embryonic karyotype abnormalities, defining the remaining 1,480 cases as case group (A)” in **Results** (page 17, lines 210–216).

To clarify our case definition in the stratified analysis and provide the results when focusing exclusively on the cases with ascertained euploidy, we added the descriptions as; “We note that case groups (A), (B), and (C) contained uRPL cases with unknown karyotypes to retain sample size for the stratified analysis. To focus exclusively on euploid and aneuploid miscarriage, we defined the 203 cases with confirmed embryonic euploidy as case group (D) and the 124 cases with confirmed embryonic aneuploidy as case group (E). We performed a GWAS of case group (D) and (E), finding no genome-wide significant association (Supplementary Fig. 2d,e). In these settings, HLA-C*12:02–HLA-B*52:01–HLA-DRB1*15:02 haplotype did not reach statistical significance due to the reduced sample size ($P = 0.16$ for (D) and $P = 0.30$ for (E); Fig. 3).” in **Results** (page 18, lines 236–243) and revised **Fig. 3** as indicated below.

(revised) Figure 3. Associations of HLA-C*12:02–HLA-B*52:01–HLA-DRB1*15:02 haplotype in the stratified analysis

Odds ratios of HLA-C*12:02–HLA-B*52:01–HLA-DRB1*15:02 haplotype in the stratified analysis are shown. The marker size is proportional to the case sample size. Error bars indicate the 95% CI. case group (A), uRPL cases with confirmed embryonic euploid or undetermined karyotype; case group (B), ANA(+) uRPL cases with confirmed embryonic euploid or undetermined karyotype; case group (C), autoantibody and hypothyroidism(-) uRPL cases with confirmed embryonic euploid or undetermined karyotype; case group (D), uRPL cases with confirmed embryonic euploid; case group (E), uRPL cases with confirmed embryonic aneuploid.

Third, would it be possible to validate the CDH11 CNV with a different method, either with a different bioinformatic tool or with qPCR?

Response:

Thank you for the thoughtful comment. We agree with you that technical validation of CNVs by orthogonal methods is an important point. Nevertheless, we would like to ask you to kindly note that rare CNVs detected by large-scale SNP array data generally have a unique challenge in validation using other methods, such as PCR-related methods. We would like first to clarify in detail what the detected association signal represents. The CNV calling method we used in this study, HI-CNV, leverages SNP array probe intensity signals and haplotype-sharing information (*i.e.*, identity-by-descent [IBD] information) to detect large contiguous intensity signal shifts accompanied by specific IBD haplotypes. Since the SNP array probes are designed to be distributed genome-wide at intervals of more than 5-kbp, the current method only reports whether a SNP site is contained in a detected CNV region, but does not locate the exact genomic coordinates at which the CNV starts or ends with a resolution of 1-bp. This methodological feature imposes unique challenges on validation steps by orthogonal methods, such as designing primers for PCR. Moreover, this feature suggests another possibility that the detected association signal potentially represents not a simple deletion but a more complex structural variant involving shifts in the array probe intensities captured by the specific IBD haplotype shared among the cases. In such situations, the validation by other methods faces further technical challenges.

While we admit that the current methodological limitation requires future research to identify the exact structure and genomic coordinate of the rare CNVs (or more complex structural variants), the etiological role of the *CDH11* variants is highly plausible considering the following lines of evidence by the prior clinical and experimental studies:

- (i) *CDH11* missense variants were implicated in the pathogenesis of RPL by previous whole-exome sequencing work (Quintero-Ronderos P *PLoS One* 2017).
- (ii) *CDH11* mutations have been reported as a causative gene for several mendelian disorders (Anazi S *Mol Psychiatry* 2017; Taskiran EZ *AM J Med Genet A* 2017; Harms FL *Am J Med Genet A* 2018; Li D *Hum Genet* 2021).
- (iii) *CDH11* shows especially high expression in the female reproductive system at both RNA and protein levels (according to *Human Protein Atlas*).

Given the biological plausibility, while your point is well taken, we would like to report the rare variant association at *CDH11*, even if technical validation is currently challenging.

Modification:

To clarify the technical limitations the current SNP array-based CNV calling methods have, we added the descriptions as:

“Our study has some potential limitations. ... Last, the rare CNVs at *CDH11* were detected using a SNP array intensity-based method. Since the SNP array probes are designed to be distributed genome-wide at intervals of more than 5-kbp, locating the precise genomic coordinates at which the variants start or end involves technical challenges. We also note another possibility that the rare variant association potentially represents more complex structural variants rather than simple deletions.” in **Discussion** (page 24, lines 336–352).

Besides, given that the rare variant association may involve more complex structural variants, we revised the abstract by replacing the term “CNVs” with the more general term “variants” as:

“Genome-wide copy-number variation (CNV) calling demonstrates rare predicted loss-of-function (pLoF) variants of the cadherin-11 gene (*CDH11*) conferring the risk of uRPL ($P=1.3\times 10^{-4}$; OR=3.29 [95% CI: 1.78–5.76]). Our study highlights the importance of reproductive immunology and rare variants in the uRPL etiology.”

Minor comments:

-Please clarify if the karyotypes of the abortuses refer to the latest miscarriage? How many women had had miscarriages with multiple partners?

Response and modification:

Thank you for raising an important point. To clarify how the embryonic karyotypes were determined, we added the description as:

“When a missed miscarriage was diagnosed, a dilatation and curettage or manual vacuum aspiration was performed and cytogenetic analysis of products of conception was carried out. Subsequent pregnancy outcomes were followed up until May 2023 by a review of the medical records. Embryonic karyotypes were primarily examined in the latest pregnancy loss for the patients. Some cases were examined multiple times, and if both euploid and aneuploid were observed, they were classified into aneuploid miscarriage.” in **Methods** (page 25, lines 370–376).

Also, we truly appreciate your suggestion of adding a description of the number of patients with multiple partners. We admit that analyzing those female patients would enable us to efficiently focus on uRPL cases certainly caused by maternal factors. While we genuinely appreciate your insightful viewpoint, we do not have a record of that in the current dataset and would like to leave it for future work.

- Please also provide the CI-s and effect allele frequencies in the abstract.

Response and modification:

Thank you for the comment. We revised the abstract accordingly.

REVIEWERS' COMMENTS

Reviewer #1 (Remarks to the Author):

The authors have adequately addressed the concerns brought up in my previous review.

Reviewer #2 (Remarks to the Author):

The authors have been responsive to my comments and I have no further suggestions.

Response to the reviewers' comments:

Reviewer #1 (Remarks to the Author):

The authors have adequately addressed the concerns brought up in my previous review.

Response:

We sincerely thank you for your valuable help in shaping our study.

Reviewer #2 (Remarks to the Author):

The authors have been responsive to my comments and I have no further suggestions.

Response:

We are very grateful for your time and effort in reviewing our manuscript.